# Effect of Drought and Methyl Jasmonate Treatment on Primary and Secondary Isoprenoid Metabolites Derived from the MEP Pathway in the White Spruce *Picea glauca*

**DOI:** 10.3390/ijms23073838

**Published:** 2022-03-30

**Authors:** Erica Perreca, Franziska Eberl, Maricel Valeria Santoro, Louwrance Peter Wright, Axel Schmidt, Jonathan Gershenzon

**Affiliations:** 1Max Planck Institute for Chemical Ecology, 07745 Jena, Germany; feberl@ice.mpg.de (F.E.); msantoro@ice.mpg.de (M.V.S.); aschmidt@ice.mpg.de (A.S.); gershenzon@ice.mpg.de (J.G.); 2Faculty of Biological Sciences, Friedrich Schiller University, 07745 Jena, Germany; 3Zeiselhof Research Farm, Pretoria 0102, South Africa; lwright@tutanota.com

**Keywords:** biotic stress, abiotic stress, monoterpene, diterpene, carotenoid, chlorophyll, abscisic acid, jasmonic acid

## Abstract

White spruce (*Picea glauca*) emits monoterpenes that function as defensive signals and weapons after herbivore attack. We assessed the effects of drought and methyl jasmonate (MeJA) treatment, used as a proxy for herbivory, on monoterpenes and other isoprenoids in *P. glauca*. The emission of monoterpenes was significantly increased after MeJA treatment compared to the control, but drought suppressed the MeJA-induced increase. The composition of the emitted blend was altered strongly by stress, with drought increasing the proportion of oxygenated compounds and MeJA increasing the proportion of induced compounds such as linalool and (*E*)-β-ocimene. In contrast, no treatment had any significant effect on the levels of stored monoterpenes and diterpenes. Among other MEP pathway-derived isoprenoids, MeJA treatment decreased chlorophyll levels by 40%, but had no effect on carotenoids, while drought stress had no impact on either of these pigment classes. Of the three described spruce genes encoding 1-deoxy-D-xylulose-5-phosphate synthase (DXS) catalyzing the first step of the MEP pathway, the expression of only one, *DXS2B*, was affected by our treatments, being increased by MeJA and decreased by drought. These findings show the sensitivity of monoterpene emission to biotic and abiotic stress regimes, and the mediation of the response by DXS genes.

## 1. Introduction

Frequent and prolonged episodes of drought in recent decades have triggered and increased the rate of tree mortality and forest dieback throughout the world [1]. Well-documented examples from southern Europe, Canada and western North America point to the wide range of species and forest types affected [2,3,4]. Drought kills trees by causing hydraulic failure and by limiting photosynthesis, leading to carbon starvation. A reduction in photosynthesis can also increase susceptibility to biotic stresses, such as herbivores and pathogens [5], by restricting the production of constitutive or inducible chemical defenses [6,7]. However, there is conflicting information about how the biochemical and physiological changes induced by drought can impact the chemical defenses of plants [8]. Some studies provide evidence of a reduction in defenses under drought, while others show no change or even an increase in defenses [9,10,11,12]. 

Volatile organic compounds (VOCs) are an important component of chemical defenses in plants that act by repelling herbivores and by attracting herbivore enemies [13,14,15]. In both gymnosperms and angiosperms, it has been shown that VOC emission usually increases after herbivore attack [16,17,18]. The defense hormone jasmonic acid (JA) appears to play a major role in triggering this increase [17,19]. The application of exogenous JA or its volatile methyl ester methyl jasmonate (MeJA) elicits VOC responses as well [20,21,22,23,24].

The VOC composition of many gymnosperms and some angiosperms is dominated by monoterpenes, a class of compounds with over 1000 known structures [25]. These C_10_ terpenoids can be emitted from pools stored in special secretory structures, including resin ducts and secretory cavities, or from any tissue after de novo biosynthesis [26]. Such an emission can be constitutive or induced upon biotic stress [15]. Monoterpenes have been implicated in direct and indirect defense against herbivores. For example, the acyclic monoterpene alcohol linalool repels aphids from transgenic Arabidopsis [27] and attracts ichneumonid parasitic wasps to poplar trees in the field [17]. For many monoterpenes, however, clear evidence of their influence on herbivores and pathogens is still missing. Moreover, rather than a single monoterpene, the composition of the total volatile blend could have an emergent role in the interaction with other organisms [28]. Together with monoterpenes, diterpenes are also the main components of the resin in many conifer species, which is known to defend trees against herbivores and pathogens both chemically and mechanically [29,30].

The biosynthesis of monoterpenes and diterpenes begins with the synthesis of C_5_ isoprenoid units via the 2-C-methyl-D-erythritol 4-phosphate (MEP) pathway, localized in plastids. This pathway also produces C_5_ units for primary isoprenoids including carotenoids, chlorophyll side chains and various plant hormones, such as gibberellins, abscisic acid and strigolactones. The operation of the MEP pathway may be limited by abiotic stresses, such as drought, since its primary substrates come directly from photosynthesis. However, the MEP pathway has been shown to maintain substantial operation under drought by increasing the recruitment of substrate from alternative carbon sources besides photosynthesis, such as chloroplast starch deposits or stored glucose transported by the xylem [31].

The first step of the MEP pathway, catalyzed by 1-deoxy-D-xylulose 5-phosphate synthase (DXS), is often described as rate determining [32]. This enzyme was found to play an important role in regulating the metabolic flux in *P. glauca* under both moderate drought and well-watered conditions [31]. DXS gene expression has been reported to be closely positively correlated with the DXS enzyme. Thus, it is not surprising that the *DXS* expression level has a close relationship to the concentrations of isoprenoids made from MEP pathway-derived C_5_ units, such as carotenoids and chlorophylls [33,34,35].

Interestingly, there is a small gene family for *DXS* reported from several species of plants [36,37,38]. In *Picea abies*, three different *DXS* genes (*DXS1*, *DXS2A*, and *DXS2B*) encode proteins with distinct functions in primary and secondary metabolism [36,39]. *PaDXS1* was found to be constitutively expressed and not be influenced by wounding or pathogen infection, suggesting that it is mostly involved in primary metabolism. On the other hand, *PaDXS2A* was induced by chitosan and methyl salicylate application, both elicitors of anti-fungal defense responses in plants, and *PaDXS2B* was induced by treatment with methyl jasmonate and wounding, suggesting involvement in anti-herbivore defense responses. Thus, based on present knowledge, type 2 *DXS* genes are likely to be involved in the biosynthesis of defense compounds such as resin monoterpenes and diterpenes and other isoprenoids of secondary metabolism [36,38,39], but this needs to be investigated further and expanded to additional species.

In this study, we investigated how the defensive and non-defensive isoprenoid metabolites of a conifer respond to a combination of drought and methyl jasmonate (MeJA) treatment, used as a proxy for herbivore attack. *Picea glauca* (white spruce), a common tree species in North American forests, was chosen since it is becoming subjected to more frequent and intense episodes of drought [40,41]. We measured the effect of both drought and MeJA on the emission of volatile monoterpenes and other MEP pathway products, such as diterpenes, carotenoids, and chlorophylls. In order to get insight into the regulation of the MEP pathway under stress, we also evaluated the expression of different *DXS* genes. Our results indicate that drought and MeJA influence monoterpene emission through the regulation of a specific *DXS* gene.

## 2. Results

### 2.1. Drought Increases Abscisic Acid (ABA) Levels, Methyl Jasmonate Treatment Increases Jasmonic Acid (JA) Levels, and Combined Drought plus Methyl Jasmonate Treatment Is Additive

Treatment of *P. glauca* saplings with drought significantly increased levels of the hormone ABA, but jasmonate treatment, applied as methyl jasmonate (MeJA), had no significant effect (Figure 1A). ABA accumulation under drought and under combined drought + MeJA treatment was higher than the control (Figure 1A), but the interaction between the drought and MeJA treatments was not statistically significant [two-way ANOVA, drought: F_1,15_ = 34.214, *p* < 0.001; MeJA: F_1,15_ = 2.913, *p*= 0.108; drought*MeJA: F_1,15_ = 0.541, *p* = 0.473]. On the other hand, the hormone jasmonic acid (JA) was increased by the MeJA treatment, but not by the drought treatment (Figure 1B). However, here again there was no significant interaction between drought and MeJA treatments, [two-way ANOVA, drought: F_1,15_ = 3.261, *p* = 0.091; MeJA: F_1,15_ = 98.798, *p* < 0.001; drought*MeJA interaction: F_1,15_ = 0.37, *p* = 0.849] (Figure 1B). The accumulation of ABA and JA in the combined drought + MeJA treatment was thus additive of what occurred in the separately applied drought and MeJA treatments.

### 2.2. Carotenoid Content Is Stable under Drought and Methyl Jasmonate Treatments, Whereas Chlorophyll Content Decreases under Methyl Jasmonate Application

The total pool of carotenoids measured, including β-carotene, lutein, neoxanthin, and violaxanthin, was not affected by either treatment separately or in combination (Figure 2A), [two-way ANOVA, drought: F_1,15_ = 0.572, *p* = 0.461; MeJA: F_1,15_ = 3.013, *p* = 0.103; drought*MeJA interaction: F_1,15_ = 0.820, *p* = 0.380]. However, the pool size of chlorophylls a and b was significantly reduced by the MeJA treatment, though not affected by the drought treatment (Figure 2B). The combination of both treatments reflects the effect of MeJA, showing a reduction in the chlorophyll pool size compared to the control [two-way ANOVA, drought: F_1,15_ = 1.148, *p* = 0.301; MeJA: F_1,15_ = 16.831, *p*= 0.001; drought*MeJA interaction: F_1,15_ = 0.502, *p* = 0.239].

### 2.3. MeJA Induces a Large Increase in Monoterpene Emission and an Alteration in Monoterpene Composition That Is Partially Suppressed by Drought

At the first sampling, five days after drought treatment began and one day after MeJA application, monoterpene emission was significantly increased by the MeJA and the combined drought + MeJA treatment, but not by the drought treatment alone [two-way ANOVA, drought: F_1,15_ = 3.561, *p* = 0.079; MeJA: F_1,15_ = 45.722, *p* < 0.001; drought*MeJA interaction: F_1,15_ = 7.093, *p* = 0.018] (Figure 3A). The increase with respect to the control was 11-fold after MeJA treatment alone and 3.7-fold after drought + MeJA treatment.

The volatile monoterpene blend can be divided into three different groups: non-oxygenated (α-pinene, camphene, β-pinene, myrcene and limonene), oxygenated (camphor, borneol, terpinen-4-ol, α-terpineol, piperitone and bornyl acetate), and a third group referred as “ induced only “. This group consists of monoterpenes detected only as volatiles ((*E*)-β-ocimene and linalool) (Figure 4, Appendix A) that are not present in the stored pool (see below for list of stored monoterpenes), but most prominent after MeJA treatment. The blend of control trees was mostly characterized by non-oxygenated compounds, which made up 85% of all emitted monoterpenes, with 14.5% belonging to the oxygenated compound group (Figure 4A). Under drought treatment, the proportion of oxygenated compounds made up 43% (Figure 4A). After MeJA treatment, composition was dominated by non-oxygenated and induced only monoterpenes (Figure 4A). For the combined drought + MeJA treatment, monoterpene composition was intermediate between those of the drought and MeJA treatments, consisting of a higher portion of induced only compounds and oxygenated compounds compared to the control (Figure 4A).

The second sampling, which took place after 14 days of drought and 10 days after MeJA application, showed a general reduction of monoterpene emission. For example, monoterpene emission in control trees was reduced by 10-fold compared to the first sampling (Figure 3B). Drought treatment had no significant effect on monoterpene emission (Figure 3B) [two-way ANOVA, drought: F_1,15_ = 0.015, *p* = 0.905]. After MeJA treatment, trees emitted higher amounts of monoterpenes compared to the control (Figure 3B) [two-way ANOVA, MeJA: F_1,15_ = 18.834, *p* < 0.01]. Emission after MeJA application was lower in the second sampling compared to the first, which means that the induction effect of MeJA declined over time. The drought + MeJA combined treatment reduced the emission of monoterpenes compared to the MeJA treatment alone, even though the interaction was statistically not significant [two-way ANOVA, drought*MeJA interaction: F_1,15_ = 3.384, *p* = 0.086] (Figure 3B).

As at the first sampling point, non-oxygenated compounds (82%) dominated emission from control trees at the second sampling point (Figure 4B). Under drought, oxygenated compounds increased from 10% to 16% compared to the control (Figure 4B). After MeJA treatment, the composition was dominated by non-oxygenated (69%) and oxygenated compounds (29%), with the induced-only compounds making up only 2% of the total monoterpene composition, even though their absolute emission increased by eight-fold over the control (Appendix A, Figure 4B). Drought + MeJA gave a composition with a slightly higher proportion of oxygenated compounds compared to the control (Figure 4B). Overall, the effect of the treatments on the total monoterpene composition was stronger at the first sampling than the second sampling (Figure 4, Appendix A).

### 2.4. The Pools of Stored Monoterpene and Diterpene Resin Acids in Needles Are Not Affected by Drought or MeJA

When the drought, MeJA, combined and control treatments were compared, there was no significant difference in the content of stored monoterpenes (tricyclene, α-pinene, camphene, β-pinene, myrcene, limonene, (+)-4-carene, terpinolene, camphor, borneol, α-terpineol, piperitone and bornyl acetate). There was also no significant difference in the content of stored diterpene resin acids (sandaracopimaric acid, isopimaric acid, levopimaric acid, dehydroabietic acid, abietic acid and neoabietic acid) (Figure 5A,B).

### 2.5. The Number of Resin Ducts in Needles Is Not Affected by Drought or MeJA

In a previous report [20], MeJA treatment was shown to induce traumatic resin ducts in *P. abies* stems, but not in needles. In order to test if the MeJA treatment applied in this study had an effect on the number of resin ducts in *P. glauca* needles, resin ducts were observed by microscopy. However, the number of the resin ducts was not changed by any of the treatments compared to the control (Figure 6).

### 2.6. Of the DXS Genes, Only Expression of DXS2B Is Influenced upon Drought and MeJA Treatments

The expression of genes encoding the first enzyme of the MEP pathway of terpene biosynthesis, 1-deoxy-D-xylulose-5-phosphate synthase (DXS) was measured after the drought and MeJA treatments. DXS1 and DXS2A gene expression were not significantly affected by any of the three treatments (Table 1) [two-way ANOVA, DXS1: drought: F_1,15_ = 1.563, *p* = 0.247; MeJA: F_1,15_ = 4,433, *p* = 0.0.68; drought*MeJA interaction: F_1,15_ =1.383, *p* = 0.273; DXS2A: drought: F_1,15_ = 3.389, *p* = 0.103; MeJA: F_1,15_ = 1.591, *p* = 0.243; drought*MeJA interaction: F_1,15_ = 1.570, *p* = 0.246]. In contrast, DXS2B gene expression was reduced significantly by drought stress (nearly 10-fold) and significantly increased (over 500-fold) by MeJA treatment.

The combination of drought and MeJA resulted in an intermediate DXS2B expression (about 15-fold above control), which was also higher than in the drought treatment, but considerably lower than in the MeJA treatment (Table 1), [two-way ANOVA, drought: F_1,15_ = 28.368, *p* = 0.001; MeJA: F_1,15_ = 70.434, *p* < 0.001; drought*MeJA interaction: F_1,15_ =16.825, *p* = 0.003].

## 3. Discussion

Understanding how a plant responds to natural situations requires experiments that combine biotic and abiotic stresses. Here we investigated the separate and combined effects of drought and MeJA treatments (a proxy for herbivory) on the primary and secondary isoprenoid metabolism of *P. glauca* needles.

### 3.1. Drought Has Little Effect on Primary or Secondary Isoprenoid Metabolism but Alters the Composition of the Volatile Monoterpene Blend

After 14 days of drought treatment, the content of the phytohormone abscisic acid (ABA) increased significantly in *P. glauca* needles compared to the needles of well-watered control trees (Figure 1A). Since ABA has a critical role in mediating plant physiological response to drought [42], this demonstrates that the drought treatment was effective. The immediate effect of ABA accumulation is a reduction of stomatal conductance in order to prevent transpiration and thereby water loss from the leaf. However, this response causes a reduction in photosynthesis and hence leads to a decline in fixed carbon acquisition [5,43].

Drought in *P. glauca* had no effect on the pools of various isoprenoid metabolites. The levels of the photosynthetic pigments, the chlorophylls and the carotenoids did not change significantly, as reported in a previous study [31]. In addition, the pools of stored monoterpenes and diterpenes were stable. This might be explained by the findings of previous work, which found that the formation of stored terpenoids is restricted to a brief period at the beginning of the growing season with accumulation unchanged thereafter [44,45]. Furthermore, drought had no significant effect on the monoterpene emission at both samplings.

The isoprenoid metabolism changes documented are positively related to changes in the expression of the *DXS* genes, which encode the rate-determining step of the MEP pathway. The lack of change in the expression of *DXS1* (which encodes an enzyme involved in primary metabolism [36,39]) under drought is consistent with the fact that the levels of the primary metabolites, carotenoids and chlorophylls, were unchanged by this stress. The decreased expression of *DXS2B* (which encodes an enzyme involved in secondary metabolism [36,39]) under drought might reduce the carbon allocation to the terpenoids of secondary metabolism.

The lack of change in monoterpene emission under drought at both sampling points is in contrast to the reduced *DXS2B* gene expression under these conditions. Perhaps monoterpene emission relies on existing supplies of the DXS protein without needing additional transcription. Furthermore, emission may not require much de novo biosynthesis in *P. glauca*, where daily emission is one order of magnitude lower than the amount of stored monoterpenes. It is also possible that emission occurs from stored monoterpene pools. However, drought has variable effects on monoterpene emission from conifers that store these compounds [46]. Further studies of monoterpene biosynthesis and volatilization, including their relation to storage structures and stomata are needed to understand what controls emission in *P. glauca* and other conifers under various environmental conditions. A method to distinguish between emission from stored pools and emission from de novo biosynthesis would be helpful in this regard.

Despite the lack of any significant influence of drought on the monoterpene emission rate at both sampling points in this study, this stress triggered a change in monoterpene composition. There was an increase of the portion of oxygenated monoterpenes under drought including alcohols, ketones and esters, as compared to the dominance of olefins under controlled, well-watered conditions. This pattern has also been observed in other plant species under drought [12,47,48], suggesting that the higher emission of oxygenated monoterpenes is a frequent chemical signal of drought stress. However, whether the emission itself plays a role in biotic interactions, i.e., as defense against herbivores or microorganisms, or intra-or inter-plant communication, needs further investigation.

Since ABA is also an isoprenoid product via the cleavage of carotenoid precursors, the increased formation of this hormone under drought might also feedback on the control of DXS [49]. The impact of ABA on DXS expression and activity is still poorly known.

### 3.2. Application of MeJA Leads to Over a 10-Fold Increase in Monoterpene Emission via Increased Expression of a Specific DXS Gene

Using MeJA treatment, we observed a dramatic increase in monoterpene emission following application consistent with results on another *Picea* species [17] and other woody species such as poplar [12]. JA is well known as a key regulator in the defense response to herbivores in all plants, including trees. Thus, it is not surprising that it stimulates the emission of volatiles which may act in trees as direct and indirect defenses against herbivorous insects as well as intra-plant and inter-plant defense signals. In conifers, many previous reports document the role of volatiles as defensive weapons against herbivores and pathogens [50,51,52]. However, the role of volatiles in defense signaling still needs to be investigated.

MeJA also caused an over 500-fold increase in the transcript levels of *DXS2B*, as in other species upon herbivory, wounding or MeJA treatment [36,39,53]. In *Catharanthus roseus* the expression of *DXS* type 2, but not *DXS* type 1, is induced by the overexpression of ORCA3, a jasmonate-responsive transcription factor that induces monoterpene-indole alkaloid biosynthesis [38]. Increased expression of the DXS2B enzyme, as opposed to other DXS isoforms, should provide more substrate for the de novo biosynthesis of the terpenoids of secondary metabolism, leading to increased monoterpene emission. The emission of unique monoterpenes (the “induced only” category) after MeJA treatment confirms that de novo biosynthesis occurred, since these compounds were not found in the extracts of needles as part of the stored resin. Furthermore, since the needles were not subjected to any mechanical damage by the MeJA treatment, the resin ducts with their content of stored terpenes must have remained intact. Thus, the terpene volatiles released, including oxygenated, non-oxygenated and induced only compounds, were likely all products of de novo biosynthesis regardless of category.

The emission of monoterpenes upon MeJA application decreased significantly over time, being much lower 10 days post-application (second sampling) compared to one day after application (first sampling). Curiously, *DX2B* gene expression was still high at the second sampling time, indicating that its transcript level did not directly control the rate of monoterpene biosynthesis. Decreased biosynthesis despite the high *DXS2B* transcript levels could arise from a number of factors, such as a reduction in substrate supply to the MEP pathway, a reduction in DXS enzyme activity, or a reduction in the activity of monoterpene synthases, enzymes whose activity is also affected by the application of MeJA [21].

In contrast to volatile monoterpenes, the levels of stored monoterpenes and diterpenes did not change significantly upon MeJA application. Therefore, increasing *DXS2B* expression only affected the biosynthesis of volatile monoterpenes. This suggests separate compartments for the biosynthesis of stored and volatile monoterpenes in the needles. The cells of the resin ducts, which synthesize the stored monoterpenes, were likely not metabolically active at the time of this experiment [44,45]. Hence, the increased *DXB2B* expression may have occurred only in cells participating in volatile monoterpene biosynthesis, possibly the mesophyll cells. Although MeJA application was shown to increase the number of resin ducts in the stem of *P. abies* [20], there was no effect on the number of resin ducts in needles in our study. The resin ducts in needles may not be inducible since these organs lack meristematic regions like the cambium to differentiate new cells.

Among primary isoprenoid metabolites, the carotenoid pool was stable after MeJA treatment, consistent with the stable expression of the *DXS1* gene. However, MeJA treatment did cause a decline in chlorophyll content, which may result from increased chlorophyll degradation, as shown in Arabidopsis [54].

### 3.3. Drought Suppresses MeJA Induction of Monoterpene Emission via Decreased DXS2B Expression

For the phytohormone, carotenoid and chlorophyll levels, we observed no significant interaction effect of drought and MeJA treatments, which means that the combination of both stresses on these primary isoprenoid metabolites corresponded to the addition of each single treatment. Each stress influenced metabolites independently and did not hinder or boost the effect of the other. For example, during combined treatment, JA and ABA accumulation increased, the carotenoid pool size was stable, and chlorophyll depletion occurred in the same way as for MeJA treatment alone.

For isoprenoid secondary metabolites, the outcome of the combined stress treatments was different. Monoterpene emission stimulated by MeJA treatment was strongly attenuated by simultaneous drought stress at both sampling times. Reduction in monoterpene emission should decrease the ability of *P. glauca* to defend itself with volatiles either as direct or indirect defenses or to communicate with volatiles as internal signals as in angiosperm [55,56] to alert other parts of the tree to herbivore attack. This might reflect a reduced priority for anti-herbivore defense responses under an abiotic stress like drought. Similar results were found in the evergreen woody plant *Camellia sinensis* where drought and MeJA inhibited the emission of several volatile compounds. [57]. On the other hand, drought increased the emission of other volatiles in this species, such as methyl salicylate [57]. Likewise, methyl salicylate emission increased from the tree *Alnus glutinosa* treated with drought plus herbivores [58]. Methyl salicylate may increase defenses against fungal pathogens and sap sucking herbivore like aphids through the activation of systemic acquired resistance (SAR) [59,60]. These findings suggest that drought might also influence defenses against certain biotic stresses via direct induction or priming.

More research on trees is needed to understand the outcomes of multiple stresses and the mechanisms mediating these outcomes, especially with regard to the role of volatile signals.

Changes in the composition of plant volatiles as well their total emission rate could affect the outcome of biotic interactions [28]. In this study, drought stress decreased the emission of the “induced only” compounds *(E)*-β-ocimene and linalool, which might cause a change in the amount of defensive weapons or defense signals. Drought stress has often been reported to increase the commitment of plants to defense [8], possibly because it raises the value of existing plant organs by increasing replacement costs.

Similar to the decrease in monoterpene emission we observed that the MeJA-stimulated increase of *DXS2B* expression was strongly attenuated by drought, being suppressed from over 500-fold to about 15-fold. These results indicate that both stresses, drought and MeJA application, mutually antagonize each other at the level of *DXS2B* regulation. However, the actual reduction in monoterpene emission under drought may be due not only to reduced *DXS2B* expression, but also to the reduction in MEP pathway flux as a result of the reduction in the supply of pathway substrates or to a general decline in tree function under drought.

Our study supports the idea that separate MEP pathways exist in plants producing different end products that are regulated by distinct DXS isoforms. In *P. glauca*, DXS1 (unchanged by stress treatments here) might provide the substrate for primary isoprenoid metabolites, such as chlorophylls, carotenoids and the isoprenoid hormones (since their levels were also unchanged by the treatments applied). Meanwhile, DXS2B likely participates in the formation of monoterpene volatiles, since its transcript level follows the trends in volatile emission, being induced by methyl jasmonate. DXS2A is also reported to be involved in producing C_5_ units for induced defensive terpenes, but responds to signals of fungal invasion and not herbivory [36]. The lack of increase in *DXS2A* expression in this study is consistent with the MeJA treatment applied that mimicked a herbivory and not fungal invasion.

The level of DXS2B activity itself could be regulated by hormone signaling, such as the cross talk between ABA and JA. Previous investigations showed that an overlap of the ABA and the JA pathways can have a synergistic or antagonistic effect. Cross talk can occur at level of the transcription factor MYC2 [61,62] or at level of the TOPLESS (TPL) co-repressor. Since ABA is an end product of the MEP pathway, metabolic flux through the pathway could also feedback on *DXS2B* expression.

## 4. Material and Method

### 4.1. Experimental Set Up

*Picea glauca* trees were obtained as seedlings from the Laurentian Forestry Centre, Quebec, Canada, in 2016, and grown for 1.5 years under controlled environmental conditions in a growth chamber. Summer conditions (six months, 16/8 h for day/night, 22 °C and PAR 1000 μmol m^2^ s^−1^) and winter conditions (three months, 8/12 h for day/night, 5 °C and PAR 200 μmol m^2^ s^−1^) were alternated in the chamber. At the time of the experiment, trees were under summer conditions and had already flushed their new growth. In total, 20 trees were used for the experiment and split into four groups with five replicates each: untreated controls, drought treatment, methyl jasmonate (MeJA) application, and the combination of drought and MeJA. MeJA application was used to simulate a herbivory one. Even though these two treatments do not have identical effects [63], application of MeJA is more convenient and reproducible and also had the advantage of keeping the anatomical structures of the needles intact for microscopic investigation. Overall, the experiment took 14 days.

The experiment was performed outside of the Max Planck Institute for Chemical Ecology in Jena, in the period between July and August 2018. In order to avoid irrigation by rain, the trees were arranged in a cage (5 m^2^) with the top and the side covered with a transparent tarp (grid film made of high-density polyethylene/polypropylene fabric). Furthermore, the trees were kept in smaller cages (two to four plants/cage) in order to avoid any natural herbivory. After three weeks of acclimatization, half of the trees were subjected to drought (no irrigation), whereas the other trees were irrigated daily (Appendix A).

On the fourth day after drought application, five trees of the irrigated group and five trees from which irrigation was withheld were subjected to the application of MeJA by spraying. The application of MeJA consisted of spraying the trees with a solution of 10 mM MeJA (Sigma-Aldrich, Munich, Germany) in distilled water as described previously [21]. The other trees were sprayed with water. All of the trees were then enclosed individually with polyethylene terephthalate (PET) foil (Toppits^®^ Bratschlauch, Minden, Germany) to prevent volatile interactions.

One day after the MeJA application, volatile collections were done with all treatment groups (first sampling). For the control treatment, only the measurements of four trees were used. Ten days post-MeJA application (second sampling), volatile collections were repeated and then tissue was sampled. The youngest branch on the top of the shoot was harvested, frozen in liquid nitrogen and stored at −80 °C. The needles of this branch, which all represented the current year’s growth, were used later for the biochemical analysis. The rest of the needles were also collected and the total dry weight of all needles was used for the calculation of monoterpene emission. One tree of the drought + MeJA combined treatment showed severe senescence before the end of the experiment and therefore was not sampled at the second sampling point.

### 4.2. Phytohormone Analysis

In order to measure ABA and JA, 10 mg freeze-dried, ground leaf material was extracted with 1 mL methanol containing the following internal standards: 40 ng mL^−1^ D_6_-abscisic acid (Santa Cruz Biotechnology, Dallas, TX, USA) and 40 ng mL^−1^ D_6_-JA (HPC Standards GmbH, Cunnersdorf, Germany). The extracts were analyzed on an Agilent 1260 Infinity high-performance liquid chromatography (HPLC) system (Agilent, Santa Clara, California, USA) coupled to an API 5000 tandem mass spectrometer (AB Sciex, Framingham, MA, USA) as described previously [64]. A Zorbax Eclipse XDB-C18 column (50 × 4.6 mm, 1.8 μm) was used for the chromatographic separation with a formic acid (0.05% in water)/acetonitrile gradient (flow: 1.1 mL min^−1^). ABA and JA were detected via multiple reaction monitoring and quantified relative to the peak area of the standards.

### 4.3. Carotenoid and Chlorophyll Analysis

Chlorophylls and carotenoids were extracted with 2 mL of acetone by shaking for 6 h at 4 °C in the dark. After centrifugation for 5 min at 5000 rpm at 4 °C, 800 µL of the extract was transferred into a new light-protected tube and 200 µL of water was added. After centrifugation, the supernatant was transferred to brown glass vials for analysis on an HPLC Agilent 1100 Series with UV diode-array-detector. UV detection was set at 445 nm for the detection of carotenoids and at 650 nm for the chlorophylls. A Supelcosil column LC-18 (7.5 cm × 4.6 mm × 3 µm; Sigma Aldrich) was used to separate the pigments using an acetone (solvent A): 1 mM NaHCO_3_ (in water, solvent B) gradient with a flow rate of 1.5 mL min^−1^. The gradient started with an initial mobile phase of 65% (*v*/*v*) solvent A. Then, solvent A was increased to 90% in 12 min and to 100% in 8 min. Solvent A was then held for 2 min and then decreased again to 65% after 3 min. Quantification was done by using external standard curves prepared with authentic standards of chlorophylls and β-carotene (Santa Cruz Biotechnology) analyzed in a range from 0.1 mg mL^−1^ to 0.00625 mg mL^−1^. The quantities of lutein, neoxanthin and violaxanthin were determined by reference to the β-carotene standard curve assuming a similar response factor.

### 4.4. Monoterpene Emission Measurements

For volatile collection, trees were enclosed in PET foil bags tied around the stem. Charcoal-filtered air was pumped into the bags at a flow rate of 1.0 L min^−1^ and pumped out at a flow of 0.8 L min^−1^ through a trap packed with 20 mg Super Q adsorbent (ARS, Inc., Gainesville, FL, USA) for 1 h starting at 12:00 p.m. Preliminary experiments verified that the traps were not saturated under these conditions. Immediately after sampling, the traps were eluted with 200 μL of dichloromethane containing nonyl acetate as an internal standard (10 ng μL^−1^). The solution was then analyzed using a gas chromatographer coupled to a mass spectrometer or to a flame ionization detector (GC-MS/FID) as previously described [17].

### 4.5. Extraction and Analysis of Monoterpenes and Diterpenes

According to Martin at al. (2003) [21], fresh tissue (100 mg) was immersed in 1.5 mL of tert-butyl methyl ether containing 150 μg mL^−1^ each of isobutylbenzene and dichlorodehydroabietic acid as internal standards, and shaken for 14 h at room temperature. The ethereal extract was transferred to a fresh vial and washed with 0.3 mL of 0.1 M (NH_4_)_2_CO_3_ (pH 8.0). In order to analyze diterpene resin acids, one aliquot of the extract was methylated by adding 0.2 M N-trimethylsulfonium hydroxide in methanol (Macherey-Nagel, Düren, Germany). The rest of the extract was filtered through a Pasteur pipette column filled with silica gel (Sigma 60 Å) and anhydrous MgSO_4_, and was used for monoterpene analysis [20].

Analysis of monoterpenes was performed on a GC-MS with a Hewlett-Packard 6890 system, using a DB-WAX column (0.25 mm × 30 m, 0.25 μm, J&W Scientific, Folsom, CA, USA) and a splitless injection at a temperature of 220 °C. Helium was used as a carrier gas at a constant flow of 1 mL min^−1^. The GC was programmed with an initial oven temperature of 40 °C (3-min hold), a ramp of 5 °C min^−1^ until 80 °C, then a ramp of 5 °C min^−1^ until 200 °C, followed by a final ramp of 60 °C min^−1^ until 280 °C (4-min hold). Analysis of diterpene compounds was performed on the same GC-MS instrument fitted with the same column. One microliter of the derivatized ethereal extract was injected splitlessly with an injector temperature of 220 °C. The instrument was programmed for an initial temperature of 120 °C and increased at a rate of 1 °C min^−1^ to 150 °C, followed by 5 °C min^−1^ up to 280 °C (6 min hold). Helium was used as a carrier gas. Furthermore, a GC-FID analysis of monoterpenes and diterpenes was carried out with a Hewlett-Packard 6890 system fitted with a DB-WAX column and equipped with a flame ionization detector (FID) operated at 300 °C. The GC-FID was operated under the same conditions as described above for the GC-MS analysis. GC-FID- and GC-MS-generated peaks were integrated using Hewlett-Packard Chemstation software. Concentrations of monoterpenes and diterpenes were calculated by comparing the integrated peak area with that of the internal standard isobutylbenzene and dichlorodehydroabietic acid, respectively. The identification of terpenes was based on a comparison of retention times and mass spectra with authentic standards or with mass spectra in the Wiley library.

### 4.6. Microscopy of Resin Ducts

Single frozen needles were embedded with an optimal cutting temperature (OCT) medium (Tissue-Tek, Torrance, CA, USA) and mounted onto a cryostat (HM 525, Thermo, USA) with the specimen holder held at −20 °C. Transverse sections of 60 μm were collected on a glass slide. Digital images were taken with an inverted light microscope (Axiovert 200; Carl Zeiss) at 16× magnification.

### 4.7. Extraction of RNA and cDNA Synthesis

Extraction of RNA from needles was performed according to Schmidt & Gershenzon 2007 [65]. RNA concentration and purity were evaluated using a spectrophotometer (NanoDrop 2000c, ThermoScientific, Wilmington, DE, USA). RNA was treated with TurboDNase (ThermoFisher Scientific, https://www.thermofisher.com (accessed on 30 Novermber 2021)). Next, cDNA was prepared from 1 μg of RNA using SuperScript™ III reverse transcriptase and oligoprimers (Invitrogen, Carlsbad, CA, USA).

### 4.8. qPCR Analysis

A quantitative real-time PCR was performed with BrillantSYBR Green QPCR Master Mix (Agilent Stratagene, SantaClara, CA, USA). For the qPCR reaction, 1.0 mL of cDNA, 10 mL of SYBR Green Supermix (Bio-Rad) and 1 mL of forward and reverse primers were used. Samples were run on a Bio-Rad CFX Manager 3.1 PCR machine (Bio-Rad Laboratory, Hercules, CA, USA) in an optical 96-well plate. PCR conditions corresponded to initial incubation at 95 °C for 3 min followed by 40 cycles of amplification (95 °C for 5 s, 60 °C for 10 s). Data for the melting curves were recorded at the end of cycling from 60°C to 95 °C. Transcript abundance was normalized to the abundance of the ubiquitin gene [65]. Three biological replicates, with three technical replicates per biological sample, were used. To verify that no contamination occurred, no-template water samples were prepared. Fold-change calculations were performed according to the Pfaffl method [66]. The qPCR primers employed in this study were as described for *Picea abies* [36] (Appendix A). The qPCR amplification products of the *P. glauca DXS1*, *DXS2A*, and *DXS2B* genes were sequenced, and showed 100% identity with the corresponding homologous genes in *P. abies*.

### 4.9. Statistical Analysis

For all statistical analyses, we used SPSS 17.0 (SPSS, Chicago, IL, USA). All data were checked for statistical assumptions, i.e., normal distribution and homoscedasticity, and analyzed by using two-way-ANOVA, with drought and MeJA as independent variables. Data are presented as the mean of five biological replicates. However, for *DXS1*, *DXS2A*, and *DXS2B* expression, three biological replicates were used. When necessary, data were log transformed. In case of non-parametric data, a Kruskal-Wallis test was used. Data are presented as boxplots with median (line), interquartile range (box), minimum and maximum values (whiskers) and outliers/extreme values (circles/asterisks).

## 5. Conclusions

Drought treatment had little effect on the primary isoprenoid metabolites of the MEP pathway in needles of two-year-old white spruce (*P. glauca*) trees, and did not influence either the emission of monoterpenes or the levels of stored monoterpenes and diterpenes in needles. However, drought did suppress the emission of monoterpenes induced by MeJA, and altered the composition of the volatile blend of monoterpenes in the volatile mixture. This suggests that *P. glauca* places a lower priority on defense signaling under drought conditions. A major regulatory step of terpene biosynthesis is the first step of the MEP pathway, DXS. Since *DXS2B* expression is strongly up-regulated by MeJA treatment, and is down-regulated by drought, DXS2B seems to be an important control point in the formation of volatile terpenes.

## Figures and Tables

**Figure 1 ijms-23-03838-f001:**
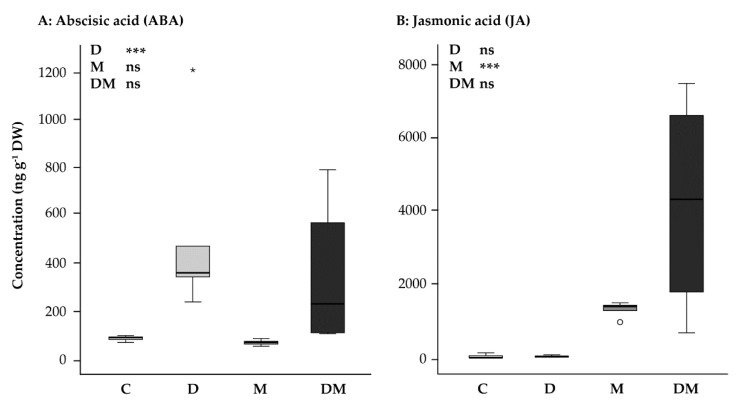
Effect of application of drought (D) and MeJA (M), separately and combined (D + M), on phytohormone levels in *Picea glauca* needles: (**A**) Abscisic acid (ABA), (**B**) Jasmonic acid (JA) at second sampling-point. Results of a two-way ANOVA are shown on the upper right: ns, not significant; ***, *p* < 0.001; white circles indicate outliers and asterisks indicate extreme values; C, control.

**Figure 2 ijms-23-03838-f002:**
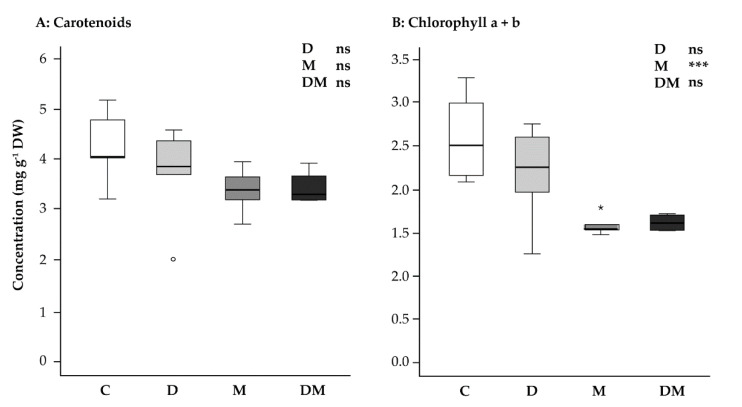
Effect of application of drought (D) and MeJA (M), separately and combined (D + M), on (**A**) carotenoids (β-carotene, lutein, violaxanthin, neoxanthin) and (**B**) chlorophyll a + chlorophyll b in *Picea glauca* needles at second sampling point. Results of a two-way ANOVA are shown at upper right. ns, not significant; ***, *p* < 0.001; white circles indicate outliers and asterisks indicate extreme values; C, control.

**Figure 3 ijms-23-03838-f003:**
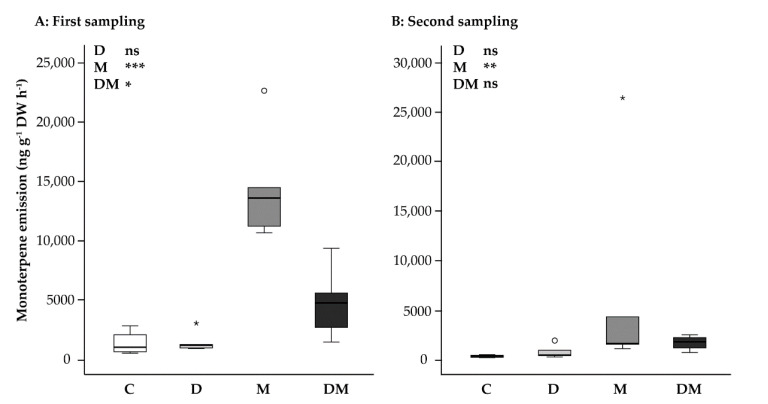
Effect of application of drought (D) and MeJA (M), separately and combined (D + M), on total monoterpene emission from *Picea glauca* needles at first at (**A**) first and (**B**) second sampling points. Boxplots show the median and dots for outliers. Shown are the results of a two-way ANOVA. ns, not significant; *, *p* < 0.05; **, *p* < 0.01; ***, *p* < 0.001; white circles indicate outliers and asterisks indicate extreme values; C, control.

**Figure 4 ijms-23-03838-f004:**
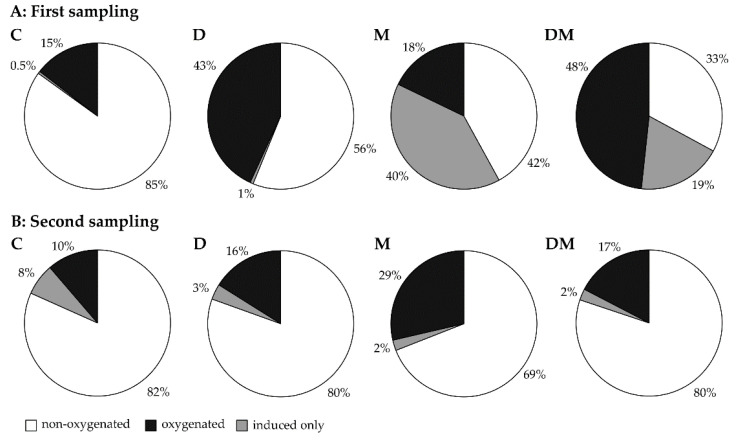
Effect of application of drought (D) and MeJA (M), drought + MeJA (DM), separately and combined (D + M), on the composition of monoterpenes of *Picea glauca* needles at first (**A**) and second (**B**) sampling points; C, control.

**Figure 5 ijms-23-03838-f005:**
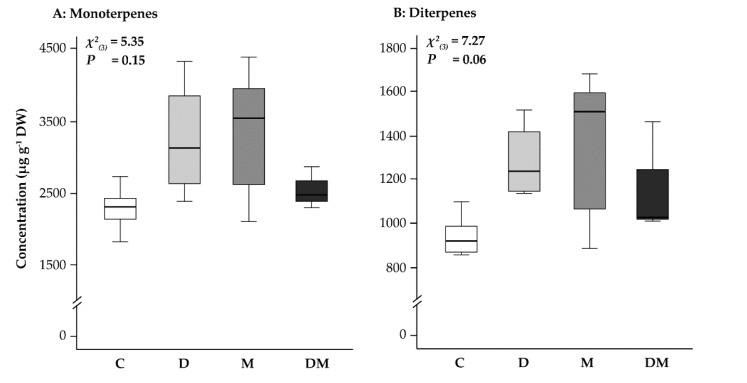
Effect of application of drought (D) and MeJA (M), separately and combined (D + M), on (**A**) stored monoterpene and (**B**) stored diterpene content at second sampling for each treatment. Results of the Kruskal-Wallis test are given on the top; C, control.

**Figure 6 ijms-23-03838-f006:**
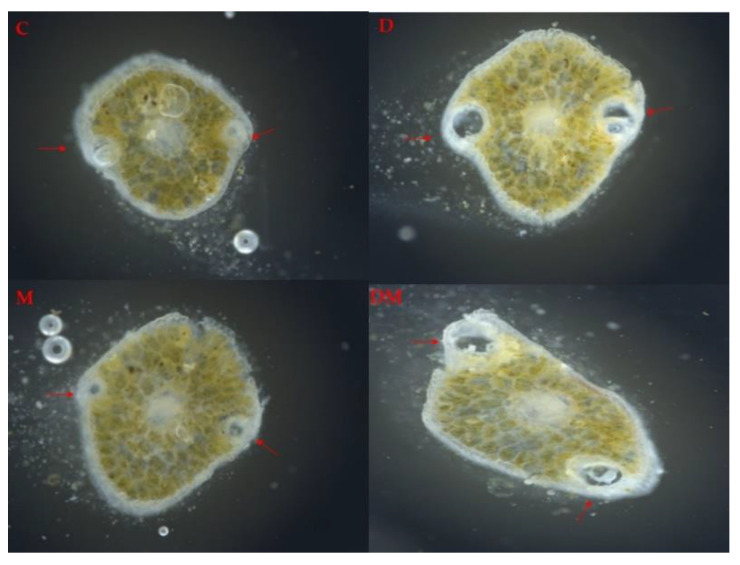
Representative *Picea glauca* needle cross sections (60 µm) under inverted light microscopy after treatment with drought (D), MeJA (M), drought + MeJA (DM) and untreated control (C). Resin ducts are indicated with arrows.

**Table 1 ijms-23-03838-t001:** Effect of application of drought (D) and MeJA (M), separately and combined (D + M), on expression of *DXS1*, *DXS2A,* and *DXS2B* at second sampling time. Data are normalized to the abundance of *ubiquitin*. Results are given as mean ± SEM, and were evaluated by two-way ANOVA. *ns,* not significant; **, *p* < 0.01; ***, *p* < 0.001; C, control.

	*DXS1*	*DXS2A*	*DXS2B*
C	0.9 ± 0.1	0.4 ± 0.3	0.9 ± 0.1
D	1.0 ± 0.2 ns	0.1 ± 0.1 ns	0.1 ± 0.0 **
M	1.2 ± 0.2 ns	1.7 ± 1.0 ns	531.8 ± 121.1 ***
DM	1.8 ± 0.4 ns	0.1 ± 0.1 ns	14.2 ± 11.9 **

## Data Availability

Not applicable.

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
