# Peer review of "Effect of Drought and Methyl Jasmonate Treatment on Primary and Secondary Isoprenoid Metabolites Derived from the MEP Pathway in the White Spruce Picea glauca"

_ijms, 2022, doi:10.3390/ijms23073838_

Round 1

Reviewer 1 Report

The manuscript studies the effects of drought, MeJA or a combination of both on primary and secondary isoprenoid metabolites in the white spruce Picea glauca. The topic is relevant and timely and relates to increased drought periods in many forest areas. It is interesting to understand how biotic stress defenses or communication cues with other organisms are affected under drought stress.

The work is based on quantifications of two hormones, ABA and JA (MeJA), isoprenoid pigments (carotenoids and chlorophylls), total monoterpenes, volatile terpene composition, stored terpenes, and expression of DXS genes under these conditions. DXS encodes a rate-limiting activity in the MEP pathway.

While MJ induces a strong increase in total amount of emitted monoterpenes, and drought does not affect quantity, the composition is distinct to each treatment, mostly in the first sampling. Drought increases proportion of oxygenated compounds, and MJ triggers the release of linalool and ocimene. Stored compounds abundance remains unchanged. The main message is then that when MJ is applied after 5 days of drought, the MJ-induced emission of monoterpenes is lowered considerably, compared to well-watered conditions. This comes along with the loss of induction of DXS2B, the main MJ-inducible isoform.

The work is mainly descriptive and fall a little short in providing some hints about the underlying mechanisms which would constitute a consistent body of novelty.

The manuscript also suffers some approximations and presentation flaws:

  1. In abstract, authors cannot claim that they addressed the effects of herbivory. MeJA mimicks some responses to herbivory, but it is a distinct stimulus. The statement should be reversed: effects of drought and MJ treatment, used as a proxi of herbivory.

Similarly, in introduction, one should not say that sapplings were treated with herbivore attack.

  1. Figure 4: ‘oxygenated/non oxygenated’ refers to structural classes whereas ‘induced’ refers to response to a stimulus. The distinction seems a bit weird at this stage, as each structural class could itself be induced or not ? what does induced only mean ? this is only known later in text. Should be clarified earlier.

  1. The methods section says MeJA was sprayed at 50 mM, an unusually high concentration. Is this correct ? Solubility of MeJA in water is < 1 g/L, and 50 mM is theoretically 11.2 g/L. So the MJ will be an oily droplet in your solution, and will not be uniformly dispersed.

  1. The observation that ‘induced’ monoterpenes are less MJ-induced in drought stressed plants is interesting, but it is not clear if this is a specific response, or due to a reduction in general fitness of the three. It would be useful to measure distinct MJ-regulated response(s) under these conditions to assess their behavior. Also, a biological readout assay would strengthen the significance of the observation.

Author Response

  1. In abstract, authors cannot claim that they addressed the effects of herbivory. MeJA mimicks some responses to herbivory, but it is a distinct stimulus. The statement should be reversed: effects of drought and MJ treatment, used as a proxi of herbivory. Similarly, in introduction, one should not say that sapplings were treated with herbivore attack.

Thanks for this comment. We agree with the reviewer!  There is enough evidence in the literature to indicate that the molecular and physiological responses to herbivory and MeJA treatment are not the same. We have now modified the abstract and introduction to show clearly that MeJA treatment was used in this study, not herbivory, but that MeJA can be considered as a proxy for herbivory.

  1. Figure 4: ‘oxygenated/non oxygenated’ refers to structural classes whereas ‘induced’ refers to response to a stimulus. The distinction seems a bit weird at this stage, as each structural class could itself be induced or not ? what does induced only mean ? this is only known later in text. Should be clarified earlier.

“Induced only” is used to refer to a group of monoterpenes that were scarcely present in the untreated plants. We have now clarified this point earlier in the introduction as suggested by the reviewer and also explain it in more detail in the results section.

  1. The methods section says MeJA was sprayed at 50 mM, an unusually high concentration. Is this correct ? Solubility of MeJA in water is < 1 g/L, and 50 mM is theoretically 11.2 g/L. So the MJ will be an oily droplet in your solution, and will not be uniformly dispersed.

Thanks to the reviewer for noting this glaring mistake! We followed the protocol of Martin et al. 2003, and the concentration we have used was 10 mM. We have now corrected this in the text.

  1. The observation that ‘induced’ monoterpenes are less MJ-induced in drought stressed plants is interesting, but it is not clear if this is a specific response, or due to a reduction in general fitness of the three. It would be useful to measure distinct MJ-regulated response(s) under these conditions to assess their behavior. Also, a biological readout assay would strengthen the significance of the observation.

We agree with the reviewer that our study does not clearly distinguish if the reduced induction of monoterpenes is due to a specific response to drought or a general decline in tree function under stress, although the reduction of DXS2B gene expression might indicate a specific response. We have now revised the discussion to indicate the possibility of this being a non-specific response to drought.

Reviewer 2 Report

  • General: it is touched on only slightly (line 290 is a good example), but I think the “herbivory” language should be eliminated from the manuscript. There is enough literature out there now showing that herbivory and MeJA application are different enough, in terms of plant molecular and physiological responses, that these terms are not interchangeable (not suggesting that is what the authors are doing, though). Simply replacing “herbivory” with “MeJA elicitation” should work. This doesn’t really change too much except for language, interpretation and usefulness of this information is still good. It’s just that if herbivory was not specifically allowed to happen, we shouldn’t use that language.
  • 37-38 (beginning to citation): I don’t think this needs to be a major change/addition, but could the authors briefly expand on this idea? Specifically, what is the conflicting information explained in the citation? This could simply be done by adding: “(e.g., blah blah)”.
  • 38-39 (after citation and comma): Unlike the above comment, I don’t know if I agree with this, especially for conifers. The pine/bark beetle literature, alone, on this topic is relatively extensive. Look up authors like Raffa, Mason, Keefover-Ring, and Erbilgin. This is especially true when MeJA application is used to represent herbivore attack. So, my recommendation would be to remove this section of the sentence (everything after the comma).
  • 43-44: There is just something about the word “dependent” in this sentence that makes me squirm ever so slightly. Unless the authors can defend/rebut the current wording (which would not offend me in the slightest is they could), perhaps reword this sentence to emphasize the role of JA? E.g., “JA appears to play a major role in the increases in VOC emission upon herbivore damage/attack”?
  • 63-65: Could the authors give a quick example of where plants alternatively pull carbon from, perhaps in the “(e.g., blah blah)” format? When the authors say “besides photosynthesis”, does that mean sugars directly from the photosynthesis assembly line or sugars and stored starch? Just some clarification on that.
  • 69-72: Perhaps a better way to make this connection is to say something like: gene expression has been found to be well and positively correlated with enzyme activity and MEP pathway products”? It’s a bit bumpy as currently written.
  • 83-84: How about: “... but this needs to be investigated further and expanded to additional species”. Perhaps not exactly that but something similar?
  • 122-123: A bit wordy, how about: “... not effected by either treatment separately or in combination...”?
  • Table 1: I believe DSX2A should be DSX1 in the first column?
  • 239-240: Another language issues. ABA doesn’t represent anything, it is the thing. And phytohormone interactions are complex, we can’t think about them in terms of “this hormone for that function” anymore. How about: “since ABA is critical in mediating plant physiological responses to drought...”?
  • 253: Was correlation analysis done? That word has a specific statistical meaning. How about “related [positively/negatively?] to” instead of “correlated with”?
  • 260-263: I’m having a hard time with this concept. I believe I understand what the hypothesis is, but if this enzyme is the rate-limiting step and there is a tight relationship between gene expression, enzyme activity, and product formation, how could a reduction in DSX2B expression lead to the diversion of resources into primary metabolism and faster turnover of carotenoids? No increase in DSX1 expression means that enzyme activity also wouldn’t change, which means carotenoid production would be no faster than it otherwise would. Help me out. I’m certainly no genius so I freely admit I could just be missing something...
  • 295-296: I suppose this is neither here not there (unless the authors want to throw in another sentence and citation), but I’m pretty confident that the literature on intra-/inter-plant VOC defense signaling in conifers is incredibly weak. That is to say, conifers do not appear to use VOC’s as defense signals within the same tree or between trees of the same conifer species.
  • 346-349: See above comment. Double-check me on that, but I am pretty sure that the evidence that conifers actually use VOCs in defense signaling is incredibly weak.
  • 352: “cases”?
  • 353-357: Hmm, I may not include this as an example as MeSA is a volatile hormone like MeJA, not a monoterpene. Different pathways, signaling, just a different animal altogether.
  • 357-358: Is there actually any evidence of an abiotic stress priming a tree to respond to a biotic stress or is this hypothesis from the authors? Do these citations (51-54) actually provide evidence that abiotic stress primes a plant to “successfully” respond to a biotic challenge? Or do these citations just show that MeSA emissions increase with drought, and the authors are suggesting that this increase in MeSA emission can benefit the plant? Because the evidence that trees can handle multiple stresses, abiotic or biotic, is incredibly weak.
  • 401-402: See general comment at top.
  • 423-425: What year neeldles were used? Were these all current or past year needles?
  • 430: ABA already defined
  • 531: I understand that a Kruskal-Wallis test can only be used to assess one-way interactions. How can a Kruskal-Wallis test be used here? I was wondering this when looking at Figure 5.

Author Response

General: it is touched on only slightly (line 290 is a good example), but I think the “herbivory” language should be eliminated from the manuscript. There is enough literature out there now showing that herbivory and MeJA application are different enough, in terms of plant molecular and physiological responses, that these terms are not interchangeable (not suggesting that is what the authors are doing, though). Simply replacing “herbivory” with “MeJA elicitation” should work. This doesn’t really change too much except for language, interpretation and usefulness of this information is still good. It’s just that if herbivory was not specifically allowed to happen, we shouldn’t use that language

We agree with the reviewer. There is enough evidence in the literature to indicate that the molecular and physiological responses to herbivory and MeJA treatment are not identical. We have now modified the text in various places to make it clear that our experiments used MeJA, not herbivory, as a treatment, but that MeJA was a proxy for herbivory.

37-38 (beginning to citation): I don’t think this needs to be a major change/addition, but could the authors briefly expand on this idea? Specifically, what is the conflicting information explained in the citation? This could simply be done by adding: “(e.g., blah blah)”.

We have now added more details about the conflicting information in the literature and more citations.

38-39 (after citation and comma): Unlike the above comment, I don’t know if I agree with this, especially for conifers. The pine/bark beetle literature, alone, on this topic is relatively extensive. Look up authors like Raffa, Mason, Keefover-Ring, and Erbilgin. This is especially true when MeJA application is used to represent herbivore attack. So, my recommendation would be to remove this section of the sentence (everything after the comma).

We have followed the reviewer’s suggestion and removed the sentence.

43-44: There is just something about the word “dependent” in this sentence that makes me squirm ever so slightly. Unless the authors can defend/rebut the current wording (which would not offend me in the slightest is they could), perhaps reword this sentence to emphasize the role of JA? E.g., “JA appears to play a major role in the increases in VOC emission upon herbivore damage/attack”?

We have now replaced the sentence as indicated.

63-65: Could the authors give a quick example of where plants alternatively pull carbon from, perhaps in the “(e.g., blah blah)” format? When the authors say “besides photosynthesis”, does that mean sugars directly from the photosynthesis assembly line or sugars and stored starch? Just some clarification on that.

We have now clarified this point and hope it is clear.

69-72: Perhaps a better way to make this connection is to say something like: gene expression has been found to be well and positively correlated with enzyme activity and MEP pathway products”? It’s a bit bumpy as currently written.

We have now revised the sentence according to the reviewer’s suggestions.

83-84: How about: “... but this needs to be investigated further and expanded to additional species”. Perhaps not exactly that but something similar?

We have now replaced the sentence according to the reviewer’s suggestions.

122-123: A bit wordy, how about: “... not effected by either treatment separately or in combination...”?

We have now changed the sentence as suggested by the reviewer.

Table 1: I believe DSX2A should be DSX1 in the first column?

Yes! Thanks for catching this mistake!

239-240: Another language issues. ABA doesn’t represent anything, it is the thing. And phytohormone interactions are complex, we can’t think about them in terms of “this hormone for that function” anymore. How about: “since ABA is critical in mediating plant physiological responses to drought...”?

We agree with the reviewer and have now changed the sentence.

253: Was correlation analysis done? That word has a specific statistical meaning. How about “related [positively/negatively?] to” instead of “correlated with”?

Changed.

260-263: I’m having a hard time with this concept. I believe I understand what the hypothesis is, but if this enzyme is the rate-limiting step and there is a tight relationship between gene expression, enzyme activity, and product formation, how could a reduction in DSX2B expression lead to the diversion of resources into primary metabolism and faster turnover of carotenoids? No increase in DSX1 expression means that enzyme activity also wouldn’t change, which means carotenoid production would be no faster than it otherwise would. Help me out. I’m certainly no genius so I freely admit I could just be missing something…

We agree with the reviewer that our argument is difficult to follow and a little too speculative. We have now modified the text to focus on how the decrease in DXS2B expression reduced allocation to secondary metabolism.

295-296: I suppose this is neither here not there (unless the authors want to throw in another sentence and citation), but I’m pretty confident that the literature on intra-/inter-plant VOC defense signaling in conifers is incredibly weak. That is to say, conifers do not appear to use VOC’s as defense signals within the same tree or between trees of the same conifer species.

The reviewer is right that the literature about the role of VOCs as defense signals in conifers is weak. We have now revised the text to indicate this point.

346-349: See above comment. Double-check me on that, but I am pretty sure that the evidence that conifers actually use VOCs in defense signaling is incredibly weak.

We have now revised this statement in light of the reviewer’s comment. The evidence for VOC involvement in tree defense signaling is weak for conifers and other gymnosperms, but strong for angiosperms.

352: “cases”?

We have now corrected this mistake in language usage.

353-357: Hmm, I may not include this as an example as MeSA is a volatile hormone like MeJA, not a monoterpene. Different pathways, signaling, just a different animal altogether.

The reviewer is correct that MeSA can properly be considered a volatile hormone, but we do not think it is a “different animal” than the monoterpenes in this context. Given the recent reports on emitted monoterpenes acting as both within and between plant signals, the roles of MeSA and monoterpenes may be closer than once thought, although this remains to be documented in conifers as the reviewer correctly indicated above. We have now cited more of this recent literature, including a review (55,56), in the discussion.

357-358: Is there actually any evidence of an abiotic stress priming a tree to respond to a biotic stress or is this hypothesis from the authors? Do these citations (51-54) actually provide evidence that abiotic stress primes a plant to “successfully” respond to a biotic challenge? Or do these citations just show that MeSA emissions increase with drought, and the authors are suggesting that this increase in MeSA emission can benefit the plant? Because the evidence that trees can handle multiple stresses, abiotic or biotic, is incredibly weak.

There is indeed some evidence of abiotic stress priming in trees. The authors of two of the references cited suggested that drought was involved in priming responses to the subsequent biotic stress. We have simply passed on these suggestions. However, the reviewer is correct in being critical since the evidence may not yet meet rigorous experimental standards for priming, and so we have weakened the conclusions and have now added a sentence about the need for further research.

401-402: See general comment at top.

We have now rewritten this sentence to emphasize that though our experimental treatments used MeJA to simulate herbivory, this is not an exact replication of natural herbivore feeding.  

423-425: What year needles were used? Were these all current or past year needles?

All needles used were from current year growth, and we have now added this information to the text.

430: ABA already defined

We have now used just the abbreviation for ABA and for JA as well.

531: I understand that a Kruskal-Wallis test can only be used to assess one-way interactions. How can a Kruskal-Wallis test be used here? I was wondering this when looking at Figure 5.

To our knowledge, the Kruskal-Wallis-test is the most suitable non-parametric test to evaluate more than two independent samples. Alternative non-parametric statistical tests, such as Friedmann-, Mann-Whithey- or Wilcoxon-rank-sum-test are used for dependent groups or two groups only, which is both not the case for our data. We tried to reach the necessary statistical assumptions for parametric tests by transformation, which was not possible with our data set though, so we believe that the Kruskal-Wallis-test was the correct decision for the statistical analysis. However, we are open for other suggestions.